# Aronofsky's *Black Swan* as a Postmodern Fairy Tale: Mirroring a Narcissistic Society

**Margarete Johanna Landwehr** 

Department of Languages and Cultures, College of Arts and Humanities, West Chester University, West Chester, PA 19382, USA; mlandwehr@wcupa.edu

**Abstract:** Based on the plot of Swan Lake, Black Swan depicts an ingenue's metamorphosis into a woman and a prima ballerina that contains a fairy-tale plot in which a naïve heroine overcomes enemies and obstacles in order to achieve success and sexual maturity. Unlike a traditional fairy tale, this cinematic tale concludes with death and the clear distinctions between good and evil, helper and adversary and reality vs. fantasy are fluid. As in many fairy tales, the film criticizes the values of its era, namely, the narcissistic aspects of contemporary society with its excessive worship of youth, beauty and celebrity, and its most pernicious results—escape into fantasy and insanity, aggressive rivalry, violence, and self-destruction.

**Keywords:** Black Swan; Swan Lake; metamorphosis; narcissism; postmodern fairy tale



## 1. Introduction

In an article in the *New York Times*, Salman Rushdie (2021) describes the power of fairy tales. Although they depict places of "beautiful impossibility," they contain truths about human nature that are still relevant and powerful today:

The stories that made me fall in love with literature in the first place were tales full of beautiful impossibility, which were not true but by being not true told the truth, often more beautifully and memorably than stories that relied on being true. Those stories didn't have to happen once upon a time either. They could happen right now. Yesterday, today or the day after tomorrow. . . . Yet, their power endures; and it does so, I believe, because for all their cargo of monsters and magic, these stories are entirely truthful about human nature (even when in the form of anthropomorphic animals). All human life is here, brave and cowardly, honorable and dishonorable, straight-talking and conniving . . . (Rushdie 2021).

In short, fairy tales constitute insightful explorations of the full range of human traits—the ugly and beautiful, kindness and cruelty, love and murder, good and evil. These revelations about human relationships are often cloaked in fantastic or animal figures—the jealous witch, the fairy godmother, the seductive wolf, the handsome prince, and benevolent fairies. Passed down to generations through oral tales, they often contain wisdom on how to cope with life's challenges. This wisdom seems particularly relevant during a once-in-a-century pandemic. As Rushdie (2021) observes: "the stories ask the greatest and most enduring question of literature: How do ordinary people respond to the arrival in their lives of the extraordinary? And they answer: Sometimes we don't do so well, but at other times we find resources within ourselves that we did not know we possessed, and so we rise to the challenge, we overcome the monster, Beowulf kills Grendel and Grendel's more fearsome mother as well, Red Riding Hood kills the wolf, or Beauty finds the love within the beast and then he is beastly no more. And that is ordinary magic, human magic, the true wonder of the wonder tale.

Ballets, plays, and films sometimes serve as contemporary fairy tales as they are communal stories that we experience collectively in a theater and from which we sometimes learn lessons such as how to overcome obstacles and deal with adversaries. These communal art forms, like fairy tales, can also serve as a form of social critique by exposing

society's shortcomings and injustice through portrayals of villains or evil enemies who are punished.

*Black Swan*, a cinematic fairy tale, mixes fantasy with reality by depicting its story entirely through the perspective of an increasingly unhinged protagonist, the ballerina Nina. As in a fairy tale, her metamorphosis from naïve ingénue to a woman and prima ballerina occurs through her confronting many adversaries—her overbearing mother, a devious rival, a haunting predecessor, and a demanding ballet director. Nina's female rivals serve as her *Doppelgänger* or double figures, common in traditional and literary fairy tales. Yet, instead of the fairy tale happy ending, the film contains a dark conclusion, which portrays a society in which success can come at a high price: Nina's madness and demise depict her final achievement, her stunning premiere performance, in a questionable light. Common in postmodern fairy tales, this shocking reversal exposes and condemns a contemporary world in which the death of a beautiful, successful woman is all too frequent.

After discussing the swan motif as a symbol of the relationship between sexuality and violence, a prominent theme in the film, and the narrative sources of *Swan Lake* in Section 2, I will analyze three aspects of this cinematic fairy tale: the metamorphosis of the heroine from child to adult, a common theme in traditional fairy tales (Section 3); psychological and mythic interpretations of the "anti-fairy tale" aspects of the film, in particular, the ambiguous death scene at the film's conclusion, (Section 4); and the role of the fairy tale as social critique, in this case, of a narcissistic society (Section 5).

## 2. The Swan Motif and Sources for *Swan Lake*

Tchaikovsky's masterpiece, which premiered at the Bolshoi Theater in Moscow in 1877, has many distinguished predecessors in mythology, art, poetry, folklore and German Romantic literature in which the swan motif serves as an archetypal symbol of sexuality. Since the Greek myth in which Zeus, disguised as a swan, ravishes Leda, the swan has symbolized the potentially savage aspects of sexuality. Helen of Troy, Leda's daughter with Zeus and the King of Sparta's wife, embodies this intertwining of beauty with brutality. Helen's loveliness bewitches Paris, who kidnaps her from Sparta and triggers the Greeks' invasion of Troy. Thus, the face that launched a thousand ships, the symbol of ultimate beauty, precipitates a war and the destruction of a civilization.

This myth served as a symbol of sexuality primarily in Italian Renaissance painting. Based on a brief account of Ovid's Latin narrative poem *Metamorphoses*, which translates into "changes of shape" and constitutes a collection of tales of love and transformation, these artworks portray a woman in the act of copulation with a swan, a socially acceptable pictorial substitute for human sexual intercourse. Thus, the swan serves as a symbol of disguised human sexuality. Famous artistic depictions of this theme include copies of lost paintings by da Vinci and Michelangelo, a Roman marble statue, a painting by Correggio, and Cezanne's *Leda and the Swan*. Poems by Ruben Dario, Ronsard, and William Butler Yeats also depict this theme. Yeats succinctly links the rape, an act of violence, but also of creation, the engendering of Helen, with destruction, Troy's devastation and the murder of the Greek war hero Agamemnon by his power-hungry wife Clytemnestra upon his return from the Trojan War: "A shudder in the loins engenders there. The broken wall, the burning roof and tower And Agamemnon dead."[1]

Two probable sources of *Swan Lake's* libretto portray this intertwining of beauty with death, creativity with destruction.[2] Tchaikovsky's acquaintances recalled his keen interest in the life of the "Swan King," the Bavarian Wittelsbach King Ludwig II, who is also known as the "Fairy Tale King." Ludwig II's tragic life had been marked by the sign of the swan and may have served as the prototype for the dreamer Prince Siegfried of *Swan Lake*. Ludwig's famous castle in the Alps, *Neuschwanstein*, or "New Swan Stone," was built near the *Schwanensee* or Swan Lake. Ludwig had a grotto built on the grounds of his exquisite, neo-Roccoco style palace Linderhof for performances of his favorite opera, Wagner's *Lohengrin* whose central symbol is a swan. It was claimed that Ludwig, like Odette, committed suicide by drowning, but many believe that he was murdered.[3]

Swan Lake's libretto may be based on a literary fairy tale *The Stolen Veil (Der Geraubte Schleier)* by Johann Musaeus, a German Romantic writer. Its heroine Zoe of Naxos, a descendent of Leda and the swan, is transformed once a year into a swan in order to travel to a magical pond in Germany that bestows upon the bather eternal beauty and youth. While in the pond, she and her companions turn into nymphs. Benno, a hermit, tells the knight Friedbert, his companion, the story of his forbidden love for Zoe, a married Greek princess, his imprisonment and escape to Germany, and his settling near the pond in order to catch a glimpse of her. Later, we learn, the jealous husband who had imprisoned Benno for his love of Zoe, no longer allowed his wife to travel to the pond and her beauty withered away, because the lovers were kept apart. This intertwining of beauty and tragic love constitute the story's central theme.[4]

### 3. A Cinematic Fairy Tale: The *Doppelgänger's* Role in a Metamorphosis

Darren Aronofsky's film, which premiered at the 67th Venice International Film Festival in 2010, contains the *Doppelgänger* (double figure) motif and related themes of the potentially destructive nature of love through betrayal taken from the ballet and from a screenplay he had been revising for a film *The Understudy*. The latter drew upon Dostoevsky's story *The Double* and the melodrama *All About Eve*, which portrays the intense competition between an older actress (Betty Davis) and her younger professional and romantic rival (Anne Baxter). While working on a revision of *The Understudy*, he conceived the central idea for *Black Swan* (Dollar 2010).

Set in contemporary New York, this psychological thriller depicts the sexual and professional maturation of Nina Sayers, who strives for the coveted dual role of Odette/Odile, in *Swan Lake*. Nina's initial attempts to dance perfectly reveal her obsessive striving for the prima ballerina role, which excludes living a full life as a young adult. Two women, her domineering mother Erica and her devious rival Lily, pull Nina in opposite directions. If the former attempts to overly protect Nina, as symbolized in her child-like bedroom of pinks and whites, then Lily introduces Nina to the big-city life of sex, drugs, alcohol and men, as depicted in the red-lit underground club that she and Nina visit. Both women, and, to a lesser extent, Beth, Nina's predecessor, serve as her *Doppelgängers* or alter egos, a figure that often appears in folk tales and literary fairy tales and appears as a jealous stepmother/witch, a rival, or an adversary. These adversarial figures often depict the hero/heroine's universal, psychological struggles during the maturation process such as attempts to gain autonomy as portrayed in the mother/daughter conflict (*Snow White*) or to satisfy sexual desire in a fulfilling adult life as seen in romantic rivalry (*Cinderella*).

In his classic essay "The Uncanny," Sigmund Freud ([1970] 1982, p. 97) discusses the *Doppelgänger*, a prominent figure in German fairytales and Romantic literary works such as E.T.A Hoffmann's *The Devil's Elixir*. In such tales, the protagonist may resemble another character and often identifies with him/her. Freud cites Otto Rank's study of the literary motif of the *Doppelgänger* as a mirror image or shadow-self of the hero/heroine (Freud [1970] 1982, p. 256). Freud adds that the creation of a double can serve as a defense against the obliteration of the self and evolved from "*uneingeschraänkten Selbstliebe*," "unlimited self-love" that evolves from narcissism, which dominates the psychic life of the child and "primitive" (wo)man (258). In severe pathological cases, this second self is split off from the psyche and often projected onto another (258–59). Freud's ideas about the relationship between the Doppelgänger figure and narcissism will help to illuminate Nina's psychological development. However, Jungian theory, which focuses on universal myths and archetypes that occur frequently in fairy tales, will provide additional insight into Nina's evolution both in the context of fairy tale motifs and the psychological processes they portray. In particular, the Jungian psychoanalyst Marie-Louise von Franz, who worked closely with Jung from 1934 until his death in 1961 and who published widely on dreams and fairy tales, links the hero/ine's progress in a fairy tale with universal psychic growth. As von Franz (1987) states in *Shadow and Evil in Fairytales* (12), " . . . fairy tales mirror the most basic psychological structures of man [and woman] to a greater extent than myths

and literary products. As Jung once said, when you study fairy tales, you can study the anatomy of man [sic]." Unlike epics such as the Babylonian *Gilgamesh* or the ancient Greek *Odyssey* or myths of Athena or Zeus, which are embedded in their respective civilizations, "the fairy tale can migrate better for it is so elementary and reduced to its basic structural elements that it appeals to everybody (von Franz 1987, p. 12)."

Aronofsky's melodrama incorporates several folktale tropes including a conflict between a naïve protagonist and an evil adversary, the destructive envy of a mother towards her daughter, and the heroine's metamorphosis from child to mature woman. In Morphology of the Folktale, Vladimir Propp ([1968] 1979, p. 79) calls these adversaries villains who struggle with the protagonist. But, these villains/*Doppelgängers* ultimately and paradoxically, also serve, as "helpers" as it is through her struggle with these adversaries, particularly with her mother and Lily, that Nina gains maturation, called by Propp transfiguration. The competition between Nina and her rival Lily, for the coveted prima ballerina role calls to mind the rivalry between Cinderella and her jealous stepsisters. Erica Sayers' envy of her daughter's success mirrors the vanity and malevolence of Snow White's and Cinderella's stepmothers. Finally, Nina's metamorphosis from a child into an adult reflects the fairy-tale protagonist's dangerous journey, a metaphor for the maturation process that entails overcoming obstacles, rivals, and/or enemies.

Both the ballet and the film's early scenes, like many fairy tales, contain an essential dichotomy of innocence/evil and white/black. The film focuses on the competition between the naive, vulnerable Nina (Natalie Portman) and the shrewd, seductive Lily (Mila Kunis), a morally ambiguous figure, and Nina's rival for the coveted role of the Swan Queen. Their off-stage rivalry mirrors the on-stage one between Odette and Odile for the prince's heart. Odette (Nina), the white swan, embodies purity and grace, whereas the evil Odile (Lily), the black swan and daughter of the malicious sorcerer Von Rothbart (Red Beard), serves as Odette's dark *Doppelgänger*. Rothbart has enchanted Odette, a princess, into a swan, who regains her human form at night. Only the love of a faithful man can break the spell. (In a similar vein, the seductive ballet director Thomas, who is transformed into Rothbart in one sexual scene, encourages Nina to discover her sexuality in order to break through her paralyzing inhibitions and discover her creative potential.) While hunting, Prince Siegfried discovers Odette's transformation from a swan into a ravishing woman. He falls in love with her and invites her to a royal ball. Disguised as Odette, Odile attends the ball and tricks Siegfried into breaking his vow. Mistaking Odile for Odette, he declares his love for her. Because of his betrayal, Odette cannot be freed from her enchantment. In despair, she throws herself into a lake and the prince follows her; their death destroys Rothbart's powers and they ascend into heaven.

As folklorist Barbara Fass Leavy (1994, pp. 218–19) notes in *In Search of the Swan Maiden: A Narrative on Folklore and Gender*, in animal-bride fairy tales, associations among black=animal=nature=evil are quite common. In most mythologies, darkness is equated with evil, sex, and destruction. The dichotomy between virtue and evil, true and false brides, is also common in folklore. The fair, blond woman is sexually innocent, whereas her black counterpart embodies "depraved sexuality as well as generalized evil" (Fass Leavy 1994, p. 220). The sensual Lily wears black, whereas, early in the film, the waif-like Nina's white and pink clothing suggest her innocence. These colors dominate her bedroom with its stuffed animals and dolls. The toys, colors, and musical jewelry box with its twirling ballerina underscore Nina's childlike nature. Yet, the butterflies in the bedroom wallpaper foreshadow her maturation into an adult.

Lily, who plays a significant role in Nina's maturation, is her *Doppelgänger* (Goldenberg 2013, p. 112). Lily initiates Nina's discovery of her sexuality by enticing her away from a quiet evening with her overprotective mother to a wild night of drinking, drugs, dancing, and men in a dark, underground club that suggest a witches' Sabbath or an ancient bacchanalia. Nina's putting on Lily's seductive black lingerie top has the traits of an initiation scene, a rite of passage. The hellish-red light that dominates this scene and completes the red, white and black color trilogy common in fairy tales such as Snow White

contrasts with the whites and pinks of Nina's bedroom and resembles an underworld of taboo (unconscious?) desires, an appropriate setting for Nina's initiation into her sexuality. Lily, who reveals a pair of black wings on her back and whose name may allude to Lilith, the dark angel (or demon) from Hebrew mythology, makes sexual overtures to Nina in the cab ride home and they appear to have a sexual encounter in Nina's bedroom. This experience arouses Nina's sexual curiosity and transforms her. Her metamorphosis culminates in her sensual performance of the black swan Odile.

Lily's ambiguous role as friend and rival manifests itself in her morally questionable overtures towards Nina, which put Nina's prima ballerina status in jeopardy.[5] For example, Lily's invitation to the club causes Nina to oversleep the next morning and almost miss her crucial rehearsal as the Swan Queen. When Nina arrives at the dance hall, she discovers that Lily is serving as her stand-in. Thus, Lily's behavior is morally ambiguous, as it is unclear to viewers (and Nina) whether her invitation to a night of debauchery constitutes a friendly gesture of reconciliation or a calculated plan to provide an opportunity for Lily to usurp Nina's role as the Swan Queen (or both).

On a symbolic level, Lily, as Nina's Doppelgänger, embodies her *shadow*, that consists of unacknowledged aspects of one's personality, the unlived life, and that frequently appears in fairy tales.[6] The shadow serves as a metaphor that represents "unknown or little-known attributes and qualities of the ego" (von Franz 1964, p. 168). In dreams and myths, the shadow appears as a person of the same sex as the dreamer (von Franz 1991, p. 35) and "usually contains values that are needed by consciousness, but that exist in a form that makes it difficult to integrate them into one's life (von Franz 1991, p. 36)." Lily represents the uninhibited passion, ruthless ambition, and taboo-breaking freedom that Nina has repressed, but eventually expresses, in part, as a result of her influence. (This shadow motif also constitutes a key aspect in the performance of the Swan Queen, as the ballerina, who usually dances both the Odette and Odile roles, is instructed to dance them differently in order to show that "the evil and pure swans are at the same time different and essentially similar (Fass Leavy 1994, p. 217)"[7]). The shadow can appear during outbursts of rage, impulsive or inadvertent acts, or in dreams (von Franz 1991, p. 35). Thus, Nina's dream of dancing the role of the black swan embodies her repressed sexuality, reveals unconscious desires, and foreshadows her artistic success.

Lily not only appears to viewers as Nina's Doppelgänger, but Nina also projects her own taboo feelings—her sexual drive and her ambition—onto Lily. Both Freudians and Jungians designate an individual's perceiving a repressed aspect of the self in another as a projection: "if people observe their own unconscious tendencies in other people, this is called a projection (von Franz 1991, p. 37)." Jung described this shadow as an inferiority that individuals may wish to "escape," by "looking for everything dark, inferior and culpable in others (Jung 1970, p. 203)". As William Miller (1991, p. 39) states, "Projection is an unconscious mechanism that is employed whenever a trait or characteristic of our personality that has no relation to consciousness becomes activated. As a result of the unconscious projection, we observe and react to this unrecognized personal trait in other people. We see in them something that is part of ourselves, but which we fail to see in ourselves." Furthermore, " . . . anytime our response to another person involves excessive emotion or overreaction, we can be sure that something unconscious has been prodded and is being activated (Miller 1991, p. 40). In a similar vein, Freud states in "The Uncanny," that any emotion that is repressed can mutate, when repressed, into fear (Freud [1970] 1982, p. 263). As Nina's repressed sexuality and ambition come to the surface, she becomes suspicious, even paranoid of Lily's intentions, which is, in part, justified, and lashes out at her in anger, which suggests that Nina's increasingly intense battle with Lily mirrors an internal conflict. Moreover, in numerous scenes, such as the sexual encounter between Lily and Nina and Nina's violent dressing room "murder" of Lily during the night of her debut performance, Lily's face morphs into Nina's, which creates what I shall call cognitive ambiguity as viewers aren't certain what is "real" and what is part of Nina's disturbed and disturbing imagination. This morphing of faces from one to the other and cognitive

ambiguity between fantasy and reality intimates that Lily serves as a projection of Nina's repressed self, her shadow. Spectators' observing events through Nina's eyes blurs the distinction between reality and fantasy.

*Black Swan* also depicts a fairytale mother/daughter rivalry. Nina's possessive mother Erica (Barbara Hershey), a former ballerina who gave up her career to have a child, demonstrates ambivalence towards Nina's success, which may indicate jealousy towards her reminiscent of the evil machinations of Snow White's jealous stepmother, but also a classic Hollywood archetype. Erica attempts to control Nina's growth both as a woman and a ballerina. She encourages Nina's obsession with ballet, which, ultimately, causes her mental instability and paralyzing fear of imperfection. She regulates Nina's attempts to live a normal adult life. When Lily invites Nina to a night of clubbing, Erica attempts to prevent it under the benevolent excuse, in part justified, that Nina needs her rest for her practice the next day. Erica's suffocating control over Nina's body and sexual desire is evident in a scene in which Nina masturbates in bed; on the verge of climaxing, she suddenly sees her mother sleeping in a chair next to her bed and abruptly stops. However, as Jack Zipes (1989, p. 212) observes, the generational conflict in which "stepmothers habitually devise stratagems to retard the heroine's progress" inevitably concludes with the stepmother's defeat as "time triumphs, delivering the daughter to inescapable womanhood and the stepmother to aged oblivion or death." Although Erica stifles her daughter's growth towards maturity that contributes to Nina's insecurity, she eventually reaches both sexual and professional maturity.

Marie-Louise von Franz (1987, p. 178) calls such a parent the devouring mother who spoils her children "to death" and constantly interferes in their affairs. Erica's constant monitoring of Nina's eating habits and whereabouts through her excessive phone calls infantilizes her daughter, prevents her from reaching her full potential both as a ballerina and a woman. von Franz (1987, p. 177) explains that such mothers often have not realized their own creative potential and often consciously or unconsciously try to prevent others from reaching theirs:

> . . . if one does not live up to an inner possibility, then this inner possibility becomes destructive. That's why Dr. Jung also says that in a similar way one of the most wicked destructive forces, psychologically speaking, is unused creative power. . . . If someone has a creative gift and out of laziness, or for some other reason, doesn't use it, that psychic energy turns into sheer poison. That's why we often diagnose neuroses and psychotic diseases as not-lived higher possibilities. . . . Someone who has unlived creativeness tries to destroy other people's creativity and somebody who has an unlived possibility of consciousness always tried to blur or make uncertain anybody else's efforts towards consciousness.

Nina's attempts to satisfy her mother's demands invariably fail and she becomes increasingly rebellious towards her. Their battle over food and Nina's outing with Lily reveal a power struggle between mother and child. As Kim Chernin (1991, p. 57) states in "The Underside of the Mother-Daughter Relationship," a mother may resent the limitations of her life imposed upon her by motherhood and may feel envious, even competitive, with her daughter, who can make other choices. She can be viewed as the daughter's *Doppelgänger*, as she has taken the alternative route of motherhood rather than pursuing a career. The daughter's guilt over her mother's self-sacrifice and her own freedom to choose sometimes manifests itself in starvation or self-mutilation, symptoms that Nina exhibits.[8] She struggles between being the obedient child and the angry young woman yearning to break free from Erica's stranglehold. Instead her anger is turned inward and manifests itself as self-abuse, severe scratching. As John Sanford (1991, p. 59) notes, children repressing their anger towards parents in order to gain their approval could lead to a harmful split in the child's psyche between "the personality and shadow personality that is autonomous and, therefore, dangerous," that results in "the loss of contact with the vital energy that anger provides." When repressed, this energy can be destructive; when expressed, it can result in freedom. Marion Woodman (1982, p. 37) claims, "So long as she [a daughter] is

obedient to a mother—actual or internal—who unconsciously wishes to annihilate her, she is in a state of possession by a witch; she will have to differentiate herself out from that witch in order to live her own life." Although Nina's struggle concludes with success, an astonishing premiere performance, the unleashing of her repressed energy appears to have triggered her suicide.

This female rivalry between angelic ingenue vs. demonic witch not only constitutes a familiar conflict in fairy tales, but also dramatizes, as Gilbert and Gubar (2000, p. 36) state in *The Madwomen in the Attic*, "the essential but unequivocal relationship between the angel-woman and the monster woman" of western patriarchy. The competition between the two women results from this patriarchy, whose values are voiced through the male mirror. This observation applies not only to Nina's relationship to her mother, but also to her competition with her two rivals, Lily and Beth, as well.

Beth (Winona Ryder), Nina's predecessor and the ballet director's former lover, who is enraged at being replaced by Nina, serves as a final *Doppelgänger* figure as her fate prefigures Nina's downfall. She confronts Nina at a benefit gala and in a drunken rage accuses her of usurping her role by seducing the director Thomas. We later learn of Beth's subsequent nervous breakdown and suicide attempt, and view her broken body and spirit in a hospital room. While Nina visits Beth, she "sees" Beth stabbing herself with a nail file and is shocked when Beth's face transforms into her own. This scene, like many others, is a cognitively ambiguous one as viewers cannot be certain how much of what they see is reality or Nina's fantasy.[9] Beth's mental collapse and disfigurement from a suicide attempt foreshadow both Nina's psychological disintegration and death.

Thus, the characters of the contemporary ballet world in *Black Swan* depict a very different reality from that of traditional fairy tales that inspire trust and confidence, because even though the fairy tale hero (sic) feels abandoned in a dangerous world, he (or she) learns that "despite our ignorance of ultimate things, it is possible to find a secure place in the world (Lüthi 1996, p. 302)." As Lüthi (1996, p. 304) elaborates: "The fairy tale is the poetic expression of the confidence that we are secure in a world not destitute of sense, that we can adapt ourselves to it and act and live even if we cannot view or comprehend the world as a whole." The traditional fairy-tale protagonist has faith the s/he can succeed in overcoming obstacles through brains, brawn, and/or help from kind humans or animals. *Black Swan*, however, depicts a dangerous urban milieu in which Nina does not find secure footing (pun intended) and does not know whom she can trust. Her mother, Lily, Beth, and Thomas are all portrayed as morally ambiguous figures as their behavior is morally questionable, that is, viewers (and Nina) cannot be certain if their motives are self-serving, altruistic, or both. Lastly, *Black Swan* subverts another basic premise of fairy tales that "suffering can purify and strengthen" (Lüthi 1996, p. 303). Despite her victories over others and herself, Nina's final triumph, her brilliant performance, ends in death.

## 4. The Anti-Fairy Tale: Psychological and Mythic Portrayals of Death

The film's ambiguous conclusion simultaneously suggests the optimistic ending of a traditional fairy tale and a destructive one of a postmodern, anti-fairy tale. Wolfgang Mieder (2008, p. 50) agrees with Andre Jolles that anti-fairy tales (*Antimärchen*) have a tragic conclusion rather than the normal happy ending and contrasts "the perfect world of the fairy tale with sociopolitical issues, marital problems, and economic worries (Mieder 50)." In the traditional fairy-tale template, Nina becomes a mature adult, who has reached sexual liberation off-stage and the pinnacle of professional success on-stage. In this interpretation, Odette's drowning in a lake, which frees her from Rothbart's spell and from her white swan avatar, symbolizes the loss of Nina's innocence and her maturation. Thus, Odette's metamorphosis or transfiguration from a swan into a spiritual being reflects Nina's symbolic death and liberation from her former, psychologically crippled self. In a similar vein, psychoanalyst West-Leuer (2017, p. 1243) remarks that Nina experiences her death "as a liberation from her eternal compulsion to sacrifice herself and be sacrificed as a well-adjusted nice girl." Pavol Bargar (2018, p. 328) argues that Nina "is redeemed

by the beauty she is experiencing" and that the "perfection" she is referring to is "the redemptive power of beauty which transcends individuals." The alternative, more popular "anti-fairy-tale" interpretation suggests that Nina, like Odette, dies at the ballet's finale after stabbing herself with a mirror shard. Because the viewer shares Nina's perspective and the film's violent scenes are cognitively ambiguous, it is not clear whether her death is real or hallucinatory. Previous scenes of Nina's self-mutilation and attack of Lily prove to be imaginary. As Vignoles-Russell (2015, p. 33) observes, " . . . the film manages to sustain the protagonist's perspective and interiority from beginning to end with no interruption." Thus, the final scene could be understood as totally imagined by Nina.[10] However, the horrified reaction of Thomas and the ballerinas indicate that the red fluid on Nina's white costume is an actual wound.

Thus, despite its fairytale motifs, *Black Swan* lacks three of the four key traits of traditional fairy tales as cited by the folklorist Tolkien (1996, p. 271): fantasy, recovery, escape from death, and consolation. Most fairy tales offer the promise of the Great Escape, the escape from death and the consolation of a Happy Ending (Tolkien 1996, p. 284). Nina's probable death subverts the fairy tale's happy ending and the entire premise of conventional fairy tales as "wish dreams," as fantasy worlds in which "all things are possible" (Lüthi 1996, p. 297). As Helen Pilinovsky (2009, p. 138) notes, "Contemporary authors of fairy tales use their knowledge of Story against itself: the expectations engendered by the imprinted belief of the "happy ending" work against themselves to guarantee the opposite, the unhappy ending, the disappointment of the dystopia predicated upon the fatal flaw of unrealistic expectation." Aronfosky, like these contemporary authors, employs a full repertoire of fairytale motifs only to subvert the anticipation of a triumphant ending with an alternative tragic conclusion.

Whether a psychic or an actual death, Nina's striving for perfection in her dancing and her final words "It was perfect," offer a clue to her apparent suicide. Psychoanalyst West-Leuer (West-Leuer 2017, p. 1240) claims that Nina has a narcissistic personality with a "perfectionist ideal-ego" that "expects achievement, self-control, and functionality." Marion Woodman (1982, p. 52) describes this, ultimately, destructive pursuit of perfection as an obsession in *Addiction to Perfection: The Still Unravished Bride*:

The chief sign of the pursuit of perfection is obsession. Obsession occurs when all of the psychic energy, which ought to be distributed among the various parts of the personality in an attempt to harmonize them, is focused on one area of the personality, to the exclusion of everything else. Obsession is always a fixation, a freezing over of the personality so that it becomes not a living being, but something fixed, like a piece of sculpture, locked into a complex. There is always something catatonic about it, behind which is fear, which can accelerate into blind terror so that the person may become like a wild animal caught in the glare of headlights, unable to move.

Nina's obsessive practicing of her dance moves, her rigid control of her body and of her food and her continual monitoring of her body's movements in ubiquitous mirrors underscore her compulsive striving for perfection. Her entire life revolves around her goal to become the prima ballerina to the exclusion of everything else. Perhaps she is fulfilling her mother's unrealized dream of becoming a ballet star? This psychic frozenness prevents her from maturing sexually or emotionally and she remains stuck in childhood with her toys in her white and pink bedroom. It is not until her director Thomas encourages her to explore her sexuality and Lily drags her to the "underworld" club that Nina becomes aware of her repressed desires and begins exploring her body.

Furthermore, Woodman (1982, p. 13) states that this perfectionist attitude originates in a work-driven society in which individuals put on the mask (persona) of determined efficiency and follow society's "rhythms", but that their animal nature is denied:

All day the mask or persona performs with perfect efficiency, but when the job is done, these frenzied foreign rhythms continue to dominate body and Being. There is no "I" to call a halt, no strong, differentiated ego to gear down to the natural rhythms. If these natural rhythms have dropped down to total unconsciousness, *being* disappears, and the

body, like a beaten, terrified, neurotic animal attempts to persevere with the rhythm totally foreign to its nature. The wolf attitude which demands more and more and more during the day, howls I want, I want, I want at night. Society's values based on the work ethic and perfectionist standards, ambitions and goals uphold the wolf attitude in the professional jungle, but society can do nothing to feed the hungry wolf at night."

The frenzied tempo of Nina's daily routine—her constant dancing, her racing between home and theater on the constantly running underground subway, the frenetic dancing at the club—all reflect the constant, rapid pace of a metropolis that never sleeps.

The power play between Nina's striving for perfection and her animal (sexual) nature soon becomes dangerous. As Woodman (1982, p. 42) states: "Psychologically speaking, so long as conscious and unconscious are enemies, the ego experiences itself as in constant danger of death." Nina's growing paranoia and psychotic episodes signal her gradual descent into insanity. The opposing roles of white swan vs. black swan convey the wide abyss between her public persona, the controlled dancer, which is perfect for the white swan role, and her private sexual desires, which, once liberated, enhance her dancing as the black swan. The wider the abyss between the conscious self and the unconscious one, the greater the danger is. Woodman (1982, p. 33) describes the back-and-forth swinging between the two as a pendulum in which the psyche's energy swings between two poles: "The more the energy goes into one side, the more the compensation on the other . . . " If the resistance to one side is too strong, then it can suddenly transform into the other: "This sudden reversal of energy is called *enantiodromia*. It occurs when energy has been pushed too far in one direction, and suddenly switches into the resisting energy it has been struggling to overcome . . . . (Woodman 1982, p. 30)." This model of the psyche's conflict between public persona and the shadow (repressed desires, identities, etc.) clarifies Nina's struggle between her striving for perfection and her instinctual quest to express her sexual energy. In her white swan makeup, she "murders" her black swan, Lily, her shadow projection, in her dressing room. This brutally repressed self becomes "demonic" (the black swan). As Diamond (1991, pp. 184–85) states: "when we . . . suppress, deny, drug or otherwise try to exclude it [the shadow] from consciousness . . . we participate in the process of evil, potentiating the violent eruptions of anger, rage, social destructiveness and assorted psychopathologies that result from the daimonic [primal force of nature] reasserting itself with a vengeance into its most negative forms." Later Nina discovers that Lily is alive and that she has stabbed herself, which can be viewed as the perfectionist "white swan," attempting to annihilate the "shadow/black swan," but the shadow strikes back. Nina's astonishing transformation into the black swan on-stage constitutes the sudden and complete release of this repressed shadow, but also leads to the symbolic death of the innocent white swan and, perhaps, Nina's actual death. Bignall (2013, p. 133) states, "Nina's becoming was sudden and catastrophic, not piecemeal and selective. . . . her final triumph of the Black Swan could not be accomplished without her failure as the White Swan." Thus, by employing horror motifs, ubiquitous mirrors, and CGI technology to transform Portman into a black swan with death-like wings, Aronofsky cinematically recreates in horrifying images Nina's destructive psychic struggle.

Ballet dancers have confirmed that *Black Swan* captured the torment and high-stakes competition among dancers. Its portrayal of ballet as an art form that demands total submission rang true for New York City Ballet soloist Ellen Bar: "The psychological torture of being an artist felt real. The way it shows an artist being her own worst enemy" (Maslin 2011, p. 5). Jennifer Homans, a dance critic, observes that dance tends to attract those "who have a kind of discipline and dedication that can flip into obsession" (Maslin 2011, p. 5). The legendary Swan Lake is a "notorious soul crusher" (Dollar 2010, p. 26). Aronofsky states: "It's the Blanche DuBois of ballet, or the Othello— the one that takes the actor or actress over the edge. There's a famous story about a dancer in the Bolshoi who had a schizophrenic breakdown during 'Swan Lake.' There are so many stories about dancers losing their mind (Dollar 2010, p. 26)." American Ballet principal David Hallberg who danced the part of Prince Siegfried admits that he has felt "possessed"

while performing *Swan Lake:* "I have had performances like that...when it's almost oth-erworldly and you're possessed by something else. I had a performance when I was not only crying visibly but for about a week after, I was haunted by it. (Ng 2010, p. 1)." The screenplay may also have been prompted by dancers' memoirs such as Gelsey Kirkland's *Dancing on my Grave* and Margot Fonteyn's *Autobiography* in which she writes of the "terror" with which she faced every performance of *Swan Lake*.

The *Doppelgänger* figure also serves as a prominent mythic figure; reviewing its mythic significance will add another dimension of meaning to Nina's death. In his early work *The Myth of the Birth of the Hero* Otto Rank (1971) explores the hostile brother motif as a common mythological and literary theme. As Christine Downing (1991, p. 68) states in "Sisters and Brothers Casting Shadows," the brothers are often twins and that

In his later writings Rank subsumes this motif under that of the Double. The brother is seen primarily as an inner figure, an alter ego. The Double may represent the mortal or the immortal self, may be feared as an image of one's mortality or prized as signally one's imperishability. **The Double is death or the Immortal Soul** [my emphasis]. It inspires fear and love, arouses the eternal conflict between our need for likeness and desire for difference. The Double answers the need for a mirror, a shadow, a reflection.

Read as a universal myth, Lily's arrival in Nina's life heralds the appearance of her Double, which brings her death, but also immortality.[11] If Nina does indeed die, then her exquisite performance of the black swan grants her enduring fame. Rank recalls that the ancients regarded the shadow-like double as a spiritual, yet real being. The Greeks called this aspect of the self, which survives death and is active in dreams *psyche*.

Thus for Rank (1971), the relationship to an inner same-sex sibling, to a double, comes to signify relationship to one's unconscious self, one's psyche, and to both death and immortality. At its deepest it expresses our longing to let the ego die and to be united with a transcendent self. It signifies our longing to surrender to something larger than our ego (Downing 1991, pp. 68–69). Seen in the light of Rank's interpretation, Nina's death is both mortal and transcendent. In death she achieves immortality just as the white swan becomes a spiritual being that transcends her earthly form. Nina triumphs in death through her transcendent performance. In another reversal, Nina's death fades out to white, a symbol of purity and perfection (flawlessness), instead of black, the usual symbolic color for death.[12]

## 5. Fairy Tale as Social Critique: *Black Swan* as a Mirror of a Narcissistic Society

Like many recent retellings of traditional fairy tales, *Black Swan* can be read as a postmodern revision of a (literary) fairy tale, but with a dark undertone that contains a social critique of contemporary life. Kendra Marston (2015) views the film as a self-referential critique of the patriarchal Hollywood star system with "the female body as spectacle" (698). Marston (7) states that *Black Swan* serves as a cultural critique of gender relations and the place of women in the entertainment industry. She (701) cites Elisabeth's Bronfen (1992, xi) argument that "culture uses art to dream the deaths of beautiful women." Some film critics claim that *Black Swan* criticizes the vicious competition of the ballet world as an intensified and openly aggressive example of cut-throat capitalism. Although I support this interpretation and will review its merits below, I also believe that the film addresses the growing narcissistic trend in individuals and American society and the destructive potential of this trend.

Read as a social critique of late capitalism, *Black Swan* exposes the Darwinian fight for survival in New York, the capitalist and financial capital of the world. As the *Los Angeles Times* critic Kenneth Turan (2010, p. 1) observes, the film takes a "wildly melodramatic, unashamedly pulpy look at the blood sport that is New York City ballet." The ballerinas' resentment of the older prima donna Beth, whom they call "menopausal" and who, they wish, would retire so that one of them may replace her, reveals the potentially malicious rivalry in such a competitive and selective institution. In this postmodern interpretation, Nina, Erica, Lily and Beth, play not only the roles of innocent maiden, evil stepmother, and

devious rivals, familiar to readers of Grimms' folktales, but also are figures in a postmodern, cautionary tale of the high cost of professional success.

The gala party scene in which the ballet director Thomas flatters donors and flaunts his new star Nina, whom he informs of the company's precarious financial state and whom he persuades to join in the "performance," underscores the relationship between art and money: The ballet company needs a "fresh face" (Nina) and discards the old one (Beth) in order to present a vibrant, new product to its investors. Thus, we view Nina through Thomas's eyes as a potential financial asset as well as a gifted dancer. Furthermore, Beth and Nina appear as interchangeable commodities, professionally and personally. "My little princess," the affectionate term Thomas used for Beth and designates for Nina once she has attained star status refers to the fairy-tale aspect of her metamorphosis from unknown artist to superstar, but also to the interchangeability of his prima ballerinas and (potential) lovers and the transactional nature of personal relationships. The invitation to his apartment after the gala appears to be a prelude to a seduction scene (but isn't) and his passionate kiss and sexual advances towards Nina in other scenes are morally ambiguous. Does he wish to enable Nina to become a formidable dancer by encouraging her to discover her sexual energy? Does he desire Nina as his lover? Or does he crave the glory and financial boost of "discovering" or "creating" a new superstar? All three possibilities appear probable. As noted earlier, Beth's demise, her physical and mental brokenness, foreshadows Nina's insanity and suicide and appears as a morality tale of the devastating toll the capitalist work ethic can exert on its workforce, especially on young, vulnerable high achievers.[13]

Treating women as "marketable commodities" as Jack Zipes (1989, pp. 216–17) notes, occurs in traditional Romantic tales as well: "Because the heroine adapts conventional female virtues, that is, patience, sacrifice and dependency, and because she submits to patriarchal needs, she consequently receives both the prince and a guarantee of social and financial security through marriage." In *Black Swan* there is a reversal of this pattern. Instead of marriage, Thomas offers Nina a brilliant career, which requires patience, sacrifice, and dependency, but also such conventional "masculine" traits as ruthless ambition, a competitive spirit, and pursuing sexual desires. Nevertheless, Thomas as the powerful (male) director of a renowned ballet company still controls Nina's, Beth's, and Lily's professional fates.

*Black Swan* reveals not only the ravages of capitalism on the soul, but also the destructive aspects of an increasingly narcissistic society. The film's ubiquitous mirror imagery conjures up the myth of Narcissus, the youth in Greek mythology, who shuns Echo's affections, sees his own face while stooping to drink in a fountain, mistakes his reflection for a beautiful water nymph, and falls in love with himself. Believing that the "nymph" was shunning him when he tried to embrace her, Narcissus pines for her and dies. Because he is so obsessed with his own image, he cannot connect with others. As a symbol of obsessive self-absorption, Narcissus serves as a frequent allegory in literature and films about dance that reveal the cost of excessive self-admiration as being incapable of loving. The classic film *The Red Shoes* (1948), which is based on a Hans Christian Anderson fairy tale, could be viewed as a precursor to *Black Swan* as it portrays the protagonist's obsession with dance. An aspiring ballerina Vicky, who is torn between her dedication to dance and her desire to love, is forced to choose between her career and marriage. Like Nina, Vicky kills herself for her art by throwing herself out a window. Vicky, Nina, and Beth's obsessive quest for artistic perfection portrays the psychological and corporeal self-destructiveness of their obsession. Of course, the ubiquitous mirrors that Nina stares into refer to her internal conflict, but also serve as a crucial tool of her profession. As Alastair Macauley (2011, p. 1) notes, dancers perceive in mirrors "both the ideal versions of themselves they hope to show to the public as well as their own failings." Ballerinas' fixation on their bodies and movements clearly functions as a prerequisite for the near-perfect illusion that they present on stage.

The leitmotif of the mirror clearly serves as a rich symbol with layers of meaning. It refers not only to Nina's internal drama and conflicts with her Doppelgängers, Erica, Beth

and Lily, but also to the illusory world of ballet and as a self-referential allusion to film, the ultimate art form of illusory images. As mentioned earlier, Nina's struggle with her controlling mother mirrors the conflict between a vulnerable princess and her evil, older rival, usually a stepmother/queen as in *Snow White* with a mirror as its central symbol that ignites the plot. When the queen discovers that Snow White is more beautiful than she is, the older rival attempts to eliminate her. As Bruno Bettelheim (1977, pp. 202–3) observes,

The queen's consulting the mirror about her worth—i.e., beauty—repeats the ancient theme of Narcissus, who loved only himself so much that he became swallowed up by his self-love. It is the narcissistic parent who feels most threatened by her child's growing up, because that means that the parent must be aging. As long as the child remains totally dependent, he (sic) remains, as it were, part of the parent; he (or she) does not threaten the parent's narcissism. But when the child begins to mature and reaches for independence, then he (or she) is experienced as a menace by such a parent, as happens to the queen in *Snow White*.

Jack Zipes (2011, pp. 116–17) offers an additional level of meaning to the mirror imagery in *Snow White* that applies as well to *Black Swan*. He observes that the powerful hold that the mirror has over the queen (and, I would add, Nina) reveals social and cultural issues on a metaphorical level, namely, that women's value is determined by men, by what they deem as beautiful. He notes that the queen "becomes trapped in the spectacle of male illusions. Her identity and value as a woman are in a large part determined by the refraction of the mirror." In contemporary life, the camera lens, not the mirror, selects, manipulates, and, ultimately, creates, a standard of beauty. As Zipes (2011, p. 117) notes, "*Snow White* is interesting because the tale has prompted filmmakers to elaborate on the representation of beauty, why beauty is so important for women in men's eyes, and how the image and nature of beauty can be manipulated through the male gaze." This observation is particularly applicable in the entertainment industry—in film, theater and ballet—in which youthful beauty and grace constitute key traits in boosting a female performer's career. In *Black Swan*, Thomas, the only dominant male character, determines the professional trajectory of his ballerinas through his judgmental gaze. As noted, he replaces the older Beth with the younger Nina, as a fresh face to donors, who will fill the company's coffers. He also pits Lily, Nina's rival, against Nina through his encouraging and humiliating comments on their performances. His gaze from Nina to Lily while they are in recital clearly unnerves Nina as she clearly senses that he is comparing the two. In a conversation with her, Thomas contrasts Nina's too controlled performance with Lily's more exuberant and carefree dancing style that reveals the former's neurotic striving for perfection and the latter's confidant, even fearless, attitude. Thus, the director judges not only the external, physical beauty of the rivals' dancing, but also their inner dispositions and state of mind. It is only when Nina breaks free of Thomas's control, breaks out of Snow White's glass coffin, so to speak, and can express repressed anger and sexual desire that she liberates herself from the male gaze. Lily and Thomas may have served as (Propp's) helpers/villains who assisted Nina in recognizing her hidden qualities, but Nina emancipates herself. Her initiating a passionate kiss with Thomas, who is standing in the wings, at the height of her premier performance reverses the traditional Prince's kiss that awakens the sleeping Snow White or Briar Rose and underscores Nina's self-liberation. Zipes (2011, p. 133) expresses this liberation best: "Women cannot determine their beauty when they look into male determined and mass-mediated mirrors. Mirrors are not the bearers of truth. They have no authority, but, perhaps they can inform us about what we lack or what we need."

Furthermore, I believe that the mirror imagery functions as a reflection and commentary on our contemporary, narcissistic society. In order to build a foundation for my interpretation, I will turn to the film theory of the eminent film scholar Siegfried Kracauer. In his classic study of Weimar Cinema (1919–1933) *From Caligari to Hitler: A Psychological History of the German Film*, Kracauer theorized that film frequently reveals the *Weltanschauung* (worldview, attitude or mental climate) of a population in a particular time, as it is an art form that is produced and viewed collectively and, therefore, often expresses the desires of

its audience. Kracauer ([1947] 2004, p. 6) states: "What films reflect are not so much explicit credos as psychological dispositions—those deep layers of collective mentality, which extend more or less below the dimensions of consciousness." He (Kracauer [1947] 2004, p. 5) points out that films are never a product of an individual, but rather, of a group—director, screenwriter, editor, cinematographer, stage hands, etc.—and that films are made to appeal to the multitudes. Kracauer ([1947] 2004, p. 8) elaborates that the peculiar mentality of a nation is not fixed, but fluid: "To speak of the peculiar mentality of a nation by no means implies the concept of a fixed national character. The interest here lies exclusively in such dispositions or tendencies as prevail within a nation at a certain stage of development." The cinematic motifs that appear repeatedly in popular films depict collective wishes: "Popular films—or, to be more precise, popular screen motifs—can therefore be supposed to satisfy existing mass desires (Kracauer [1947] 2004, p. 5)." He elaborates: "In recording the visible world—whether current reality or an imaginary universe—films therefore provide clues to hidden mental processes (Kracauer [1947] 2004, p. 7)".

*Black Swan*, which enjoyed success at the box office and (mostly) critical acclaim, mirrors the contemporary American obsession with image. Twenge and Campbell (2009, p. 4) state in *The Narcissism Epidemic: Living in the Age of Entitlement*, which was published a year before *Black Swan* was released: "American culture's focus on self-admiration has caused a flight from reality to the land of grandiose fantasy." One key narcissistic trait is vanity, which is closely related to self-centeredness. The Narcissist Personality Inventory (NPI) test contains statements such as "I like to look at myself in the mirror" (2009, p. 142). Twenge and Campbell point to the emphasis not only on appearance, but also wealth, celebrity worship and attention seeking that mark an increasingly narcissistic society and cite statistics that show this trend accelerating. The quantifiable cultural changes include a five-fold increase in plastic surgery and cosmetic procedures in ten years (Twenge and Campbell 2009, p. 5). Narcissists have an overinflated view of their abilities, see themselves as superior, unique, or special, and believe they are better than others in social status, good looks, intelligence, and creativity. The most obvious examples of this obsession with image are the politicians, sports "heroes," business moguls, and entertainment figures who fabricate idealized images of themselves with the help of "spin doctors," agents, publicists, and media consultants. As Sandy Hotchkiss (2003, p. 181) observes, "We look on with jaded eye, knowing we are being manipulated but enjoying the show. Eventually reality gets so distorted that we (like Nina) no longer know who or what to believe. We find ourselves in a narcissistic fun house without a clue what's behind the mirrors." Nina's and viewers" disorientation in identifying reality vs. fantasy captures the chaos and confusion caused by multitudes of sometimes contradictory images and information that can be manipulated on television, the internet, social media, and film.

Which social factors contribute to increasing narcissism? Twenge and Campbell refer to reality TV shows which attract narcissists who love the limelight and which normalize narcissistic behavior such as over-competitiveness, materialism, appearance obsession, and the quest for fame (2009, p. 99). As Hotchkiss (2003, p. 13) states, "For Narcissists competition of all kinds is a way to reaffirm superiority" and adds that narcissists (like Nina) 'become compulsive in their pursuit of perfection. Along the way, they crave admiration from others." Social network sites make it possible to open the gap between fantasy and reality as users generally only show positive aspects of their life such as partying, or traveling. Desires and attractive personality traits are emphasized or exaggerated. On the internet, like ballet, the fantasy principle trumps the reality principle.

But, narcissists pay a price for their inflated self-image: "Narcissists also lack emotionally warm, caring and loving relationships with other people. Because they don't value deep relationships, narcissists have little concern for others and often manipulate and exploit others, whom they view as tools to make themselves look and feel good." Narcissists lack empathy and are also materialistic, entitled, and aggressive when insulted (Twenge and Campbell 2009, p. 30). This violent aspect of narcissism inspired the

researcher Del Paulhus to label narcissism as one neurosis of the "Dark Triad," which also includes Machievellianism (manipulativeness) and sociopathy (2009, p. 20).

*Black Swan* constitutes a dark, postmodern allegory of narcissistic self-destruction inherent in the obsessive pursuit of fame. Aronofsky's ballet world with its emphasis on appearance, performance, hyper-competition and success serves as a microcosm of an increasingly narcissistic society obsessed with image. Ballerinas' fixation on their bodies that, at times, include anorexia and bulimia, and hyper-vigilance with diet reflects our society's adulation of youthful beauty. The manic drive for a perfect performance and the intense rivalry among ambitious ballerinas for plum roles reflect the vicious race to the top in a society which glorifies celebrity and notoriety. The self-destructiveness of Nina, Beth, and Vicky warns that the price of this exclusive focus on appearance and success is loss—the loss of self and of meaningful relationships. When she loses her prestigious status as prima ballerina, the crippled Beth in her hospital bed expresses this loss when she cries out: "I am nothing."[14] The loss of her persona, her public image, drives Beth to annihilate herself as she believes that there is nothing left of her identity. As Marie-Louise von Franz (1987, p. 147) states: "In the archetypal experience of evil, evil powers are seen as a crippled human, or as a distorted thing, I think that we should therefore understand it symbolically and see in that the projection of a human psychological fact, namely, that evil entails being swept away by one-sidedness, by only one single pattern of behavior." (Sexeny 2015) Thus, the obsessive pursuit of one goal often involves neglecting other aspects of one's personality, the shadow. von Franz (1987, p. 8) states that this shadow can be personal and collective: "the collective shadow is particularly bad, because people support each other in their blindness—it is only in wars or hate for other nations, that an aspect of the collective shadow reveals itself." Yet, plays, ballet, opera, novels, and films can hold up a mirror to reveal our blind spots, the hidden, ignored and neglected aspects of ourselves to us.

## 6. Conclusions: *Black Swan* as a Postmodern Anti-Fairy Tale

In her influential study *Postmodern Fairy Tales: Gender and Narrative Strategies*, Cristina Bacchilega (1997, p. 23) states that postmodern transformations of fairy tales include "simultaneously affirming and questioning strategies . . . in a variety of critically self-reflexive moves." *Black Swan* offers this double movement of affirming and questioning values presented in its source *Swan Lake*. On the one hand, Nina's journey and demise mirrors that of her counterpart in the fairy-tale ballet "Swan Lake." Nina exemplifies the traditional fairy tale heroine as beautiful, innocent, persecuted and suffering. Like the White Swan Odette, she gains transcendence, through her transformative and unforgettable performance as the Swan Queen. If death transforms Odette from a suffering human/swan to a spiritual being, then Nina has liberated herself from her helpless child-like state into a courageous, sensual woman and superstar. However, the horror of Nina's insanity and death undercuts her transcendent transfiguration and suggests a critique of a contemporary narcissistic society in which humans may sacrifice parts of their identity, their socially undesirable traits, in their pursuit of perfect images and performances. This obsessive pursuit of perfection can result in a psychic, even actual death.

In particular, as Bacchilega (1997, p. 24) states, postmodern fairy tales retell "histories, values, and gendered figurations." Aronofsky's cinematic fairy tale transforms the story of the broken-hearted, pure Odette who dies because a man has betrayed her into a contemporary psychological thriller of a professional woman's self-destructive, pursuit of perfection. Moreover, the values glorified and vilified in these works reflect different social mores. The ballet, based on a 19th century literary fairy tale, exalts female innocence and submission (the White Swan) and criticizes female deviousness and duplicity (the Black Swan). The film, however, depicts the heroine's path to success as a transformation from a passive, innocent female victim into an assertive woman with such conventional "masculine" traits as ambition, aggression, and sexual drive. One could argue that a career woman's attempt to become "more like a man" is punished and that Nina's downfall

portrays, in an exaggerated and horrific way, the dilemma of contemporary working women in "a man's world." If one follows this argument, then, Nina is punished, like the evil, calculating queen in Snow White, for actively pursuing her goals instead of passively submitting to a prince.[15] However, the fairy tale dichotomies of good/evil, helper/adversary, male/female, and victim/villain are collapsed into one figure. Nina serves as both heroine and villain as she embodies both the White and the Black Swans and is responsible for both her success and death. As I have argued above, it is Nina's inability to integrate both her White Swan (her social persona) and her Black Swan (her shadow) into one harmonious whole that leads to her demise. Nina's dual roles as both Odette and Odile, as victim and perpetrator, also reflect the morally ambiguous roles of Thomas, Beth, and Lily as helpers/adversaries. With this moral ambiguity, the film subverts the conventional dichotomies of good/evil in traditional fairy tales as well as their happy ending.

Furthermore, *Black Swan* can be viewed as a postmodern anti-fairy tale in the same way that Lüthi (1996, p. 303) regards Kafka's Expressionistic tales as "anti-fairy tales". *Black Swan*, like Kafka's Existentialist stories, depicts a threatening world in which individuals are not secure, do not know whom to trust and may not even survive. Nina, like Kafka's Joseph K. in *The Trial*, is confronted with a nightmarish society with sinister denizens, whose motives one cannot decipher (moral ambiguity). Unlike protagonists and readers in the fairytale world, who can recognize helpers and villains, Nina and viewers cannot distinguish between the two, which contributes to spectators' sense of confusion. Moreover, as in Expressionist works, readers/viewers experience what I have called cognitive ambiguity, because the fictional events are portrayed through the protagonist's unreliable perspective. Because readers share Joseph's K's paranoid point-of-view, just as spectators share Nina's viewpoint, readers/viewers can never be certain what is actually happening in the fictional world of both works. Thus, one can call *Black Swan* an Expressionist film, a contemporary twin of the silent film *The Cabinet of Dr. Caligari* (1919) that is narrated by an inmate in an insane asylum.[16] We not only see what Nina sees, but we experience her disorientation with her as well.[17]

Nina's demise does offer us a moral lesson and meaning, key traits of traditional fairy tales. Holocaust surviver and child psychoanalyst Bruno Bettelheim (1996, p. 4) claims that fairy tales' central role is to provide meaning for suffering and cites the German dramatist Friedrich Schiller: "Deeper meaning resides in the fairy tales told to me in childhood than in the truth taught by life (*The Piccolomini*, III, 4). He (Bettelheim 1996, p. 309) claims that fairy tales deal with universal human problems and offer children valuable lessons about life: "That is exactly the message that fairy tales get across to the child in manifold form: that a struggle against severe difficulties in life is unavoidable, is an intrinsic part of human existence—but that if one does not shy away, but steadfastly meets unexpected and often unjust hardships, one masters all hardships and at the end emerges victorious (Bettelheim 1996, p. 311)." Although *Black Swan*'s final scene suggests that Nina's struggles have destroyed her, it does offer a moral: the obsessive quest for perfection can be fatal and humans can find happiness if they strive for wholeness instead of perfection.

**Funding:** This research received no external funding.

**Conflicts of Interest:** The authors declare no conflict of interest.

## Notes

[1] Praised as one of Yeat's masterpieces, his sonnet "Leda and the Swan" was cited by Camille Paglia as "the greatest poem of the twentieth century." (Paglia 2006, pp. 114–18).

[2] Some claim that the Russian fairy tale *The White Duck* may be another source for the ballet. In this story, a witch lures a king's young wife away from the palace and transforms her into a duck. The witch takes the queen's form. The queen has three ducklings that the witch kills and the queen/duck grieves their deaths with loud quaking. The king captures the duck and it turns into his wife's human form. They sprinkle magic water onto the dead ducklings who turn into children and the witch is killed. I could locate neither the fairy tale nor any scholarly reference to this story.

3     The alleged murder may have resulted from tax-payer resentment of his extravagant castles and limitless financial support of the composer Wagner's lavish lifestyle. Ludwig paid for many of these enterprises with his own money, but public perception is what matters.

4     Upon Benno's death, Friedbert leaves the hermitage, discovers three swans at the pond, and falls in love with Zoe's and her husband's youngest daughter Kalliste, whose veil he steals. The daughter tells Friedbert Zoe's tragic story; the couple returns to his hometown to marry. But, upon donning the stolen veil, the bride transforms into a swan and flies away to Naxos. Friedbert travels to Naxos where he finds Kalliste, who has forgiven him for the theft and agrees to share her life with him. German story downloaded from Spiegel online Projekt Gutenberg-DE on 12/30/10.

5     Thomas's seductive behavior towards Nina is equally ambiguous. Although he intimidates her in rehearsals, he also employs flattery and seduction to liberate herself from her drive for perfection. While urging her to "live a little," he encourages her to explore her sexuality so that she discovers the passion that would enhance her performance of the black swan role. His passionate kiss and sensual touch awaken Nina's erotic desire and suggest the sexual awakening of a Snow White kissed by her Prince as well as the advances of the predator Rothbart in *Swan Lake*. In an imaginary sexual encounter Nina witnesses between Lily and Thomas, Thomas transforms into Rothbart.

6     Julie Sexeny also describes the film as an intra-psychic drama and explains how Erica, Lily and Thomas impact Nina's development. She (54) believes that Nina's identification with Lily offers her the opportunity to explore repressed personality traits. Sexeny uses a different psychological framework, the theories of D. W. Winnicott and Jessica Benjamin. West-Leuer (2017, p. 1237) offers a Freudian interpretation of the intra-psychic drama and states that the battle between the swans is a repetition of an earlier sexual experience with caregivers. Jamie Goldenberg (2013, p. 106) views the black swan as "representing human creatureliness and mortality" and the white swan "as the defensive outcry against this condition" and interprets the film as "reflecting the human struggle to overcome limitations associated with mortal existence" (109).

7     See a review of Mikhael Baryshnikov's production of *Swan Lake* in Croce (1989) essay in "The Story of O." *The New Yorker*, May 29, pp. 105–7. Cited in Fass Leavy 217.

8     Chernin (57) writes: "We have seen how they (daughters) break down at the moment they might prosper and develop; we have observed the way they have tortured them-selves with starvation and make their bodies their enemies, the way they attack their female flesh. This futile attack upon the female flesh through which we are attempting to free ourselves from the limitations of the female role, hides a bitter warfare against the mother. . . . For what is a woman likely to attack when she cannot express her anger toward her mother? Isn't she likely in turning this anger toward herself, to direct it toward the female body she shares with her mother? In a stunning act of symbolic substitution, the daughter aims her mother-rage at her own body."

9     According to Dr. Dolores Malaspina, a professor of psychiatry, Nina's symptoms resemble those of a severely neurotic, obsessive-compulsive patient with features of borderline personality disorder who suffers from mini-psychotic episodes. Her harrowing visions are fantasies, not hallucinations. Rachel Loewy, assistant adjunct professor of psychiatry, believes that her hallucinations and fantasies suggest that she may have lost touch with reality altogether. She notes that stress can trigger a psychotic episode. See Marc Siegel's article. (Siegel 2011) Stefan Schubert discusses Nina's unstable identity.

10    Stefan Schubert analyzes the "narrative instability" of the film because of the "internal focalization" of events through Nina's point of view. Goldenberg also views Nina's death as bestowing on her a type of immortality. (Schubert 2013)

11    Goldenberg (112) states, "Otto Rank describes in *Beyond Psychology* (1958) how in 19th century literature it became common for the double to represent the mortality of the protagonist. He points to Dostoyevsky's *The Double* as well as Poe's *William Wilson* and Robert Louis Stevenson's *Dr. Jeykll and Mr. Hyde* . . . "

12    The white screen gets replaced with a black one when the credits roll. Schmiesig discusses the symbolism of colors in Grimms' fairy tales. Whiteness represents "freedom from outdoor toil and spiritual purity," (Schmiesing 2016) blackness usually indicates enchantment (225), but also can refer to disease, death and the underworld (219). Red, white and black are used in folk tales to symbolize women with red being associated with blood and fertility. Marston (703) states that white can be associated with pathology.

13    Marston (2015, p. 707) states: "The professional standing of Nina and Beth in *Black Swan*'s ballet company parallel that of the two stars in relation to the Hollywood industry, with Portman as a younger, more bankable alternative to the older star."

14    Sandy Hotchkiss (2003, p. 77) describes narcissists' all or nothing attitude: "if they are not better than, then they are worthless."

15    Of course, fairy tales contain not only passive princesses who are woken up by a prince's kiss, but also pro-active heroines such as Gretl in "Hansl and Gretl.")

16    Marston (703) notes that the chiaroscuro lighting, a technique that creates bold shadows and strong divisions between dark and light areas in the frame, was a visual style used in German Expressionist films and contributes to the film's sinister mood.

17    Vignoles-Russell discusses interiority of perspective in this film.

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
