# Peer review of "Aronofsky’s Black Swan as a Postmodern Fairy Tale: Mirroring a Narcissistic Society"

_humanities, doi:10.3390/h10030086_

Round 1

Reviewer 1 Report

This is a well-written and original contribution to the field of fairy-tale scholarship. On the whole I think it is a fine piece, though I believe there are some areas where it would benefit from more references and/or more explanation of the author's intentions.

Regarding Black Swan as anti-fairy tale, I'm inclined to agree, but I believe the author would do well to situate that discussion a bit in the context of existing scholarship on anti-fairy tales (for example, Wolfgang Mieder wrote a summary of the concept in The Greenwood Encyclopedia of Folktales & Fairy Tales edited by Donald Haase; I believe there is an updated edition of the reference work now with a slightly different title). Similarly, I believe I saw a reference to the film as a postmodern fairy tale, but lacking any mention of Cristina Bacchilega's groundbreaking book Postmodern Fairy Tales (I would highly recommend citing this source, both to show familiarity with important fairy-tale scholarship and to give the reader a bit more explanation as to why the author believes this term fits).

I would also strongly recommend bringing in more scholarship by Jack Zipes. His important book The Enchanted Screen: The Unknown History of Fairy-Tale Films is essential to cite in studies such as this one. For instance, a brief look at the index reveals multiple references to ballet (such as ballet's historical relationship to fairy-tale film, p. 36) that I would definitely encourage the author to incorporate, to help solidify conceptual links made in the article. Further, Zipes in his introduction to his edited volume Don't Bet on the Prince: Contemporary Feminist Fairy Tales in North America and England makes reference to narcissism, which might help the author build on the importance of narcissism within the analysis of the film (primarily pages 26-31).

My other main concern is that Jungian analysis is not universally accepted within every academic field, and while I think it works with this piece, as a reader I would be more likely to follow the author to these less-universally-accepted places if the author would explain what draws them to these theories in this piece. I don't need a lit review of what von Franz, etc. have written about fairy tales, but I do want to know why the author thinks this body of theoretical work is useful/interesting when attempting to interpret and illuminate the meaning of this particular text. For instance, I'm more trained in applying Freudian psychoanalysis to fairy tales and I could just as easily make a case for applying it to Black Swan than Jungian analytical psychology...so I want to know why, you know?

Anyway, please do not take my critique as indication that there is a LOT wrong with your paper. I hope that some of the references I've provided are useful; in my view, even if you choose a handful to pursue and mention, this will help your paper appear as even more valid and more grounded in existing fairy-tale scholarship, which will help readers primarily coming from fairy-tale studies be more receptive to your arguments. Overall, I found your analysis to be really compelling and intriguing, so re: the Jungian stuff, I mostly wanted more, on the "why" angle, so I could understand what caused you to reach these insights better.

Author Response

Thank you for an insightful and helpful review. I have incorporated all your suggestions into my paper with references to Mieder, Zipes, and Bacchilega. I've also incorporated Freud's discussion of the Doppelgänger in his essay "The Uncanny," which the other reviewer recommended and explained why I preferred Jung's discussion of the double figure over Freud's. In short, Jung studied myths and archetypes which occur in fairy tales. So, Jung's psychology not only explains Nina's conflicts with her double figures but also covers the mythical aspects of both Swan Lake and Black Swan. The sources you recommended really improved my essay. I've attached the revised essay below.

Reviewer 2 Report

Reader’s Report: Aronofsky’s Black Swan as a Postmodern Fairy Tale: Mirroring a Narcissistic Society

This is a good article submission on Darren Aronofsky’s film Black Swan (2010) as an “anti-folk tale,” and I recommend its publication. I have, however, a few comments, and I hope that the author finds these constructive as he or she revises the essay for publication.

It’s nice, by the way, to see the recent piece by Salman Rushdie cited here – the one with which this essay begins. That essay appeared a mere couple weeks ago in the New York Times, and it seemed to me to be a useful contribution to literary studies. I’m glad the author thought so as well.

-- The Doppelgänger motif is particularly prominent in German fairy tales and is generally associated with E.T.A. Hoffmann (with his novel The Devil’s Elixir, among other sources). Author might consider making that reference if he or she want to retain the point about Johann Karl August Musäus’s “The Stolen Veil” as having been a source for Black Swan. Musäus’s work would surely have been known by Hoffmann.

-- On that same page, I felt as though a couple of major pieces on the fairy tale were missing or un-cited. The author might mention Propp’s Morphology of the Folktale, if they are looking for ways to describe the structure of Black Swan, and then, of course, Bruno Bettelheim’s Uses of Enchantment should enter the picture. Specialists in fairy tale research will note the absence. (More on Bettleheim below).

-- On p. 3, there’s a reference to King Ludwig of Bavaria as having been insane, but the old “Mad King Ludwig” story doesn’t quite play as well today. He may have simply been gay and campy, and references to his madness come across as anachronistic. Most historians have done away with that trope by now.

-- The third part (beginning at Roman III) contains interesting material on the background of the film that I didn’t know. The more that the essay investigates the Nina and Lily relationship, the question loomed larger for me about whether we were in Nina’s mind for most of the film; is she, in other words, envisioning the similarities between herself, Lily, and Beth? Certainly, they seem to our eyes, as viewers, to be similar and the threat is that they are rivals and could replace her. But one also needs to know: from who’s perspective are we seeing, and are their appearances determined by Nina. Looking at the bibliography, I see an essay by Stephen Schubert from 2013, which seems like it might give some insight into the narrative voices at play in the film. Does it make any sense to speak of objective reality?

-- On p. 5, where the author makes reference to the Doppelgänger as a shadow: the author might go back to Freud’s “Uncanny” essay in which he refers to the film The Student of Prague (a film about a double), and describes how the double does things that the protagonist is generally afraid to do. All of that directly applies to Nina and Lily: Lily acts out things that Nina is afraid to act out, including her desired rebellion against her mother. I think Freud would be an interesting touchstone, especially because it seems to me that the entire film really has the structure of a dream.

-- One question that occurred to me throughout my reading of this essay had to do with the levels of ambiguity in the film: on the one hand, the author writes that a scene can be “ambiguous” insofar as we don’t know whether what the character sees is reality or Nina’s fantasy (on p. 6), but then at the top of p. 7, the author talk about “moral ambiguity,” and mentions that the film has a morally ambiguous conclusion. These are all different kinds of ambiguity, and I feel as though it’s important that we differentiate between these different kinds because one has to do with narrative voice, rather than how we react to a moral provocation or an uncertain conclusion. Could the author help the reader navigate these different types of ambiguity and their meanings?

-- When the author quotes a work by Diamond (Redeeming Our Devils and Demons, 1991; cited on p.8), I began to wonder whether the author actually believes in concepts such as the “shadow self,” or whether the author means to treat such contentions as discourses. When we write about Freud and Jung as having theorized the double, we generally discuss these as historical ideas, rather than as truths. Diamond, on the other hand, presents a relatively recent perspective (1990s) which treats the claims not as historical, but as true. The article’s author could be clearer on what are historical discourses and what are beliefs on his or her part.

-- If the author decides that he or she needs to make some cuts, then I think that the paragraph on p. 9 that begins with “Ballet dancers themselves have confirmed Black Swan captured the torment and high stakes competition among dancers,” etc., doesn’t give much to the argument and could be cut. I feel the same about some of the material on p. 12 about reality TV, which strikes me as not necessarily necessary. I believe the author’s point has already been made by this time, and that reality TV doesn’t need to enter into it.

-- In the essay’s final section (beginning with Roman V), I started to miss the theory by Bruno Bettelheim. If the argument is going to concern how the film reveals the ravages of capitalism on the soul, and the hazards of a narcissistic society, then we need a little more theorizing about why fairy tales are an effective source of that critique. This comes up, for sure, but Bettleheim (Uses of Enchantment) is an excellent source for this. Perhaps, where the author dives into a discussion of Siegfried Kracauer, I’d say that he or she could consider replacing this with some work on Bettleheim. I don’t know that we need Weimar cinema to make the argument here, and it may only serve to confuse matters.

Please check on the spellings of “Lily” versus “Lilly.” The error was only made once or twice, but the author might make sure that it is spelled consistently.

Overall, I think this is very interesting material. It is well researched, and readers will benefit from the close readings of the film. I think with a little work this would be perfectly fit for publication, and I will look forward to seeing the finished version in Humanities.

Author Response

Thank you for your insightful and helpful suggestions! I've attempted to incorporate all of them in my essay as well as possible in the requested short turnaround time of one week in which I obtained the suggested works and found appropriate passages. The additional works by Zipes, Bettelheim, Propp, and Freud are cited in the bibliography. I also added passages that emphasized the viewers' sharing Nina's perspective throughout the film, which creates the fluid boundary between fantasy and reality. Finally, I distinguished between the two types of ambiguity, which I've defined as "moral ambiguity" in which viewers are not certain of the characters' motivations (Are they helping or hurting Nina?) and "cognitive ambiguity" in which spectators share Nina's unreliable perspective in the film and cannot distinguish between reality (i.e.what is actually occurring in the fictional world) and Nina's fantasy. I believe that terms for aspects of the human psyche such as the shadow are historical (from Jung) terms that describe actual psychological realities. So, I agree with Diamond. I believe that incorporating the ideas of Zipes, Freud, etc. and clarifying the various ambiguities in the film have improved my essay and appreciate  your remarks.
